# Stance Phase Gait Training Post Stroke Using Simultaneous Transcranial Direct Current Stimulation and Motor Learning-Based Virtual Reality-Assisted Therapy: Protocol Development and Initial Testing

**DOI:** 10.3390/brainsci12060701

**Published:** 2022-05-28

**Authors:** Ahlam Salameh, Jessica McCabe, Margaret Skelly, Kelsey Rose Duncan, Zhengyi Chen, Curtis Tatsuoka, Marom Bikson, Elizabeth C. Hardin, Janis J. Daly, Svetlana Pundik

**Affiliations:** 1VA Northeast Ohio Healthcare System, Cleveland, OH 44106, USA; asalameh@fescenter.org (A.S.); jmccabe@fescenter.org (J.M.); margaret.skelly@va.gov (M.S.); kelsey.duncan@uhhospitals.org (K.R.D.); ehardin@fescenter.org (E.C.H.); 2Department of Neurology, School of Medicine, Case Western Reserve University, Cleveland, OH 44106, USA; 3Department of Biomedical Engineering, Case Western Reserve University, Cleveland, OH 44106, USA; 4University Hospitals Cleveland Medical Center, Cleveland, OH 44106, USA; 5Department of Population & Quantitative Health Sciences, Case Western Reserve University, Cleveland, OH 44106, USA; zxc30@case.edu (Z.C.); cmt66@case.edu (C.T.); 6Department of Biomedical Engineering, The City College of New York, New York, NY 10031, USA; bikson@ccny.cuny.edu; 7Brain Rehabilitation Research Center, Malcom Randall VA Medical Center, Gainesville, FL 32610, USA; jjd17@case.edu; 8School of Medicine, University of Florida, Gainesville, FL 32610, USA

**Keywords:** stroke, gait, physical therapy, virtual reality, VR, Transcranial Direct Current Stimulation, tDCS, brain stimulation, neurological rehabilitation

## Abstract

Gait deficits are often persistent after stroke, and current rehabilitation methods do not restore normal gait for everyone. Targeted methods of focused gait therapy that meet the individual needs of each stroke survivor are needed. Our objective was to develop and test a combination protocol of simultaneous brain stimulation and focused stance phase training for people with chronic stroke (>6 months). We combined Transcranial Direct Current Stimulation (tDCS) with targeted stance phase therapy using Virtual Reality (VR)-assisted treadmill training and overground practice. The training was guided by motor learning principles. Five users (>6 months post-stroke with stance phase gait deficits) completed 10 treatment sessions. Each session began with 30 min of VR-assisted treadmill training designed to apply motor learning (ML)-based stance phase targeted practice. During the first 15 min of the treadmill training, bihemispheric tDCS was simultaneously delivered. Immediately after, users completed 30 min of overground (ML)-based gait training. The outcomes included the feasibility of protocol administration, gait speed, Timed Up and Go (TUG), Functional Gait Assessment (FGA), paretic limb stance phase control capability, and the Fugl–Meyer for lower extremity coordination (FM_LE_). The changes in the outcome measures (except the assessments of stance phase control capability) were calculated as the difference from baseline. Statistically and clinically significant improvements were observed after 10 treatment sessions in gait speed (0.25 ± 0.11 m/s) and FGA (4.55 ± 3.08 points). Statistically significant improvements were observed in TUG (2.36 ± 3.81 s) and FM_LE_ (4.08 ± 1.82 points). A 10-session intervention combining tDCS and ML-based task-specific gait rehabilitation was feasible and produced clinically meaningful improvements in lower limb function in people with chronic gait deficits after stroke. Because only five users tested the new protocol, the results cannot be generalized to the whole population. As a contribution to the field, we developed and tested a protocol combining brain stimulation and ML-based stance phase training for individuals with chronic stance phase deficits after stroke. The protocol was feasible to administer; statistically and/or clinically significant improvements in gait function across an array of gait performance measures were observed with this relatively short treatment protocol.

## 1. Introduction

Nearly 80% of stroke survivors have initial gait impairment after stroke, of which 15–30% still endure severe gait deficits at 6 months [1,2]. Gait deficits after stroke are caused by several underlying impairments, which include decreased speed, pronounced gait asymmetry, and marked dyscoordination of the paretic limb [3]. These impairments lead to a significant decline in function, diminished quality of life, and limitations in life role participation [4]. Gait rehabilitation interventions can provide some improvement in the gait pattern, but unfortunately, current options do not restore gait function for many individuals [4,5].

Commonly observed compensatory strategies after stroke include incomplete weight transfer onto the paretic limb during the stance phase of gait and inability to maintain proper pelvis/hip/knee/ankle alignment as the center of mass progresses across the stance limb foot [4,6]. As a result, the impaired limb has a limited ability to provide the required support for normal stance phase characteristics. Therefore, focused training of the stance phase is important. Gait training interventions based on the principles of motor learning (ML) can improve the gait pattern [7,8,9,10,11,12,13,14]. Motor learning is an efficacious rehabilitation approach based on a high repetition of task-oriented exercises that can be incrementally progressed to ensure adequate challenge [15,16]. However, these successful interventions call for intensive and long-duration treatments that are not financially practical in the current health care milieu. Therefore, the development of methods to treat stance phase deficits more practically, effectively, and quickly is warranted [6,17,18].

We identified several shortcomings in current motor learning methods that would need to be addressed in order to improve the treatment outcome and shorten the treatment timeframe. In identifying the shortcomings, we reasoned that the following requirements should be met in developing a new training protocol: directly target brain function during training for the stance phase; provision of a strong motivation to transfer weight to the paretic limb during the stance phase; a safe environment to transfer weight to the paretic limb when imbalance and safety are an issue; a method to progressively increase the challenge of stance phase weight transfer onto the paretic limb in small and achievable increasements; and compatibility of any new methods with the array of well-known motor learning principles [16].

To meet these requirements, we identified two technologies. The first was brain stimulation technology that targets brain function directly. Transcranial Direct Current Stimulation (tDCS), a form of non-invasive brain stimulation, is known to alter brain excitability [19,20,21,22] and may potentially enhance traditional, peripherally directed stance phase training [23,24]. While the combination of non-invasive brain stimulation with physical therapy has been investigated for upper limb recovery after stroke [25], brain stimulation has not been well-explored in conjunction with gait training [26]. In the few available studies, there are mixed results. For example, a recent randomized controlled trial delivered tDCS prior to gait training, but there was no difference in gait speed across study groups; although, measures of brain plasticity had a greater change in the tDCS group [27]. The other several existing studies do favor the use of non-invasive brain stimulation [28,29,30,31]. However, there have been no studies that evaluated tDCS being delivered simultaneously with a focused motor learning-based gait training that targeted the stance phase for the paretic limb.

We identified a second technology that would provide the framework for delivering the therapy according to certain motor learning requirements; this framework was virtual environment gait training technology with a treadmill and a harness support system for safety. Others have demonstrated that the use of virtual reality (VR) during gait therapy produced better results than conventional therapy alone in some measures [32], and VR could be practical in the clinical setting [32,33]. The VR-based gait rehabilitation system we selected meets the motor learning requirement of providing a motivating task during stance phase training. Additionally, the VR gait training system includes a treadmill and harness safety system, which together can deliver the following motor learning requirements: repeated and consistent patterns of motor learning-based practice; progressively greater challenge in stance practice, along several modifiable practice domains; and a safe practice paradigm during stance phase practice [34].

Therefore, the purpose of this work was to develop and test an innovative combined stance phase gait training protocol that includes tDCS and VR gait training, with a treadmill and harness safety system, as well as over-ground stance phase practice. The protocol was delivered according to motor learning principles.

## 2. Materials and Methods

### 2.1. System Users

For the initial testing of the new stance phase training protocol, key inclusion criteria were age >18 years, medically and psychologically stable, Fugl-Meyer scale for lower extremity (FMLE) score >15 (normal = 34); FMLE item II Flexor synergy-ankle dorsiflexion score ≥1; endurance sufficient to complete 6-min walk test; ability to follow two-stage commands, such as shift weight to the left (leading) limb and lift the right (trailing) limb; and ability to provide informed consent. The key exclusion criteria were: more than one stroke, diagnosis of other neurological disorders, contraindications for transcranial magnetic stimulation (TMS) [35], Transcranial Direct Current Stimulation (tDCS), and Magnetic Resonance Imaging (MRI). All of the participants provided written informed consent before participation began. No participants were participating in standard care gait rehabilitation during the study. The study was approved by the medical center institutional review board (IRB; #15036-H19) and was conducted at the VA Northeast Ohio Healthcare System main building, Cleveland, OH, USA.

### 2.2. Development of the New Combined Stance Phase Training Protocol

#### 2.2.1. Transcranial Direct Current Stimulation (tDCS)

We identified a tDCS device (Soterix Medical, New York, NY, USA) for the delivery of low amplitude currents (few mA) to the brain through electrodes on the scalp (anode and cathode) [36]. The constant electric field reaching neural tissue modulates the resting membrane potentials and the membrane’s depolarization/repolarization mechanisms, thus altering cortical excitability [24]. Therefore, we hypothesized that tDCS could create an enhanced environment for the mechanisms of brain plasticity during stance phase motor learning [20,28,37,38,39]. We used a 1 × 1 bihemispheric montage using 5 cm square saline saturated sponge (10 mL per sponge) electrodes with an anode over the ipsilesional M1 region for the leg and a cathode on the homologous contralesional side to administer 2 mA tDCS [40]. We identified a stimulation intensity of 2 mA and a duration of 15 min based on the recommended parameters by the stimulator manufacturing company for the optimal bihemispheric stimulation of the motor area [41,42,43]. We identified the optimal electrode placement for the bihemispheric montage by identifying tibialis anterior (TA) hotspots of both the contralesional and ipsilesional hemisphere using a Magstim 200^2^ transcranial magnetic stimulator (Magstim Company Ltd., Wales, UK) guided with frameless stereotactic navigation (Brainsight2, Rogue Research, Inc., Montreal, QC, Canada) [44]. We placed surface electromyographic (EMG) electrodes over the belly and tendon of the TA muscles. We used electromyography (EMG) to record Motor Evoked Potentials (MEPs) in response to single-pulse TMS of the primary motor area (M1). We defined the optimal site for inducing MEP (“hotspot”) as the site that evoked 50 μV MEPs, using the lowest TMS intensity in the resting contralateral muscle reliably in 6 out of 10 trials [45]. If resting MEPs could not be elicited, then we had the participant activate the muscle at 20% of the maximum voluntary contraction, monitored with visual feedback of the EMG recording [45]; and a threshold of 200 μV MEPs in 6 out of 10 trials was used to determine the hotspot. Both paretic and non-paretic limb TA hotspots were identified in this manner, and these hotspots were used as the targets for tDCS stimulation in subsequent treatment sessions. We centered the tDCS electrodes over C1 and C2 (International 10–20 system), then made an adjustment in the anterior/posterior direction to be directly lateral to the tibialis anterior (TA) hotspot, maintaining the distance from the midsagittal line of C1 and C2 (Figure 1). We found it important to secure electrodes on the scalp with an elastic head strap to ensure good contact was maintained. We digitized the electrode locations in the neuronavigation system to replicate the positioning between the 10 sessions and for use in electric field modelling. In ‘bench testing’ the system, we found it important to secure the tDCS cables in such a manner as to prevent any downward force on the user’s head; to address this, we affixed the cables with tape to the user’s shoulder and then to the handrail of the treadmill, finding that this procedure effectively removed any potential downward force.

#### 2.2.2. Virtual Reality (VR) Technology

We identified the V-Gait virtual reality system (Motek Medical, Amsterdam, The Netherlands; Figure 1A) with a treadmill as capable of delivering some of the requirements that we identified for stance phase gait training. It was necessary to interface the V-Gait VR system with the user’s gait pattern. Therefore, we employed real-time motion tracking of the user. To acquire user movement, we applied markers on the lateral malleoli and used a passive marker motion capture system (Vicon, Oxford, UK) software to track the real-time movement and deliver the images to the VR screen (D-Flow, Motel Medical Amsterdam). The screen was 4 m wide by 3 m high with a visual angle of 90 degrees and was positioned 1.5 m from the treadmill. We used a custom-built VR application where individuals walked in a country scene while virtual obstacles (small rectangular blocks) were placed in their pathway. The user was challenged to step over the virtual obstacles. When the foot in VR space contacted an obstacle, the user received audio and visual feedback. We arranged all of the technologies (treadmill, harness safety, tDCS, motion capture, and V-Gait virtual environment) so that the virtual obstacle in the path was presented in the pathway of the non-impaired limb so that the user would be required to transfer weight to the paretic limb during stance phase. We monitored the performance of the user and modified characteristics of the obstacles to progress the difficulty of stance phase practice.

#### 2.2.3. VR system Components: Treadmill and Harness Safety System

We used the V-Gait system’s treadmill to manipulate training speed. Users trained at an initial optimal training speed, and we progressively increased gait speed as able. The treadmill had multiple safety features: a stop button within easy reach of the treating physical therapist and treadmill operator and sensors that would halt the treadmill if the individual progressed too far forward or too far back on the treadmill. We used an overhead safety harness that was fixed to the ceiling to ensure that if the individual lost his/her balance or misstepped, a fall would be arrested. The harness system also afforded the user a sense of confidence that a fall could not occur. Additionally, a trained staff member was always in a position to assist the individual during training.

Prior to enrolling users, we tested the accurate and reliable function of the tDCS, V-Gait virtual reality and treadmill system, motion capture system, and harness system technologies separately, as well as their respective interfaces with each other. Further, we repeated the practice of donning and booting up all technologies until the team was efficient in the timely set-up for a future user.

### 2.3. Construction of the Treatment Protocol of Combined tDCS, VR, Treadmill, Harness System, and Overground Gait Training

#### 2.3.1. Summary of Treatment Protocol

We constructed the protocol to include 10 gait training sessions (2x/week; 1 h per session). We administered all aspects of the protocol according to the known principles of motor learning, including the following: practice as close-to-normal joint movement coordination as possible [16] with continual progression toward normal as ability improved; high repetition of the desired coordinated joint movements [16]; attention focused on the motor task at hand [4], training specificity [6,17], which in this case entails the practice of gait stance phase movement components; and awareness and feedback [17]. Throughout the session, we targeted stance phase training on the paretic limb. Each session consisted of two phases. Phase one included 30 min of combined tDCS and treadmill walking in the VR environment (Figure 1). We administered the tDCS during the first 15 min of the phase one treadmill/VR training. In phase two, we provided 30 min of overground training. We constructed a custom home exercise program (HEP) for each user, also targeting the stance phase of gait. We provided instructions to complete the HEP on non-clinic days and until the 3-week follow-up visit.

#### 2.3.2. tDCS Electrical Field Modelling

tDCS electrical field distribution (EF; V/m) was modeled to estimate the electrical stimulation exposure in key anatomical regions. We used 1 mm^3^ isotropic voxel T1-weighted MRI scans collected at baseline to create an individualized head model for each patient. The EF was modeled for 5 × 5 cm sponge-based electrodes with a current of 2 mA at the location recorded in the neuronavigation system. In the computational model, the head was divided into several compartments such as the scalp, skull, cerebrospinal fluid (CSF), and gray and white matter. A stroke lesion was classified as CSF. Electrical conductivities were assigned to each compartment, and a finite element model was used to solve for the EF as described and validated previously [46,47,48]. The research group at Soterix Medical, NY, computed the EF model for each patient. The EF was qualitatively assessed in the whole head and quantified in the ipsilesional leg primary motor region, defined as the medial portion of the primary motor cortex. The region was defined for each patient individually by neurologists. Whole-brain EF models were computed, and the mean EF for each participant in the leg primary motor region is reported.

#### 2.3.3. VR, Treadmill, and Harness SAFETY system

In applying the VR environment, treadmill, and harness safety system, we employed a motor control hierarchy based on the principles of motor learning developed in prior work [3]. We used these principles and the hierarchy of difficulty to guide initial levels of challenge as well as treatment progression for the recovery of the coordinated components of gait [13,14]. We defined readiness for progression according to a number of performance factors, including the following: increasing accuracy of non-paretic limb object clearance during its swing phase; and maintaining coordinated paretic single limb support control during its stance phase. We conducted an in-session assessment of each user in this fashion and used assessment results to vary or hold constant the parameters of stance phase difficulty for immediate practice or for the progression of practice difficulty. These parameters included treadmill speed and parameters within the VR gait training protocol, such as obstacle thickness and height (constant or random heights presentation) and distance between obstacles. We programmed the VR system to present the user with 34 obstacles/run, which they were tasked to step over with the non-paretic limb. Users completed as many runs as they were able within the allotted 30 min treadmill session. Brief rests were offered between the runs and as needed by the user.

#### 2.3.4. Overground Gait Training

Overground gait training was used to reinforce practice in weight bearing on the paretic limb during the stance phase (Figure 1B). The training was tailored to address the specific underlying impairments of each individual. Exercises focused on weight shifting onto the paretic limb, stance phase pelvis/hip/knee/ankle control, gait component practice (e.g., practicing the sequence of initial contact to mid-stance and to late stance on the paretic limb), and overground walking. The difficulty was continuously progressed to ensure adequate challenge [3].

#### 2.3.5. Home Exercise Program

Participants were issued with an individualized HEP that reinforced in-clinic training and was continuously progressed to ensure adequate challenge (Figure 1C). The customized HEP consisted of exercises to promote stability in the affected limb during weight bearing. The exercises emphasized lateral weight shift onto the affected limb, forward weight shift, and pelvis/hip/knee/ankle coordination as expected in the stance phase. We instructed the users to practice their individualized HEP for at least 1 h/day for a minimum of 50 repetitions/exercises. During the in-clinic visits, we queried users regarding HEP compliance.

### 2.4. Outcome Measures

#### 2.4.1. Feasibility

We evaluated the feasibility of administering the protocol to chronic stroke survivors according to the following: user (stroke survivors) recruitment; retention for the planned duration of the testing time; at each session, user preparation and donning time of equipment; reported comfort of the system; capability to wear and use the technologies; user endurance of the technologies for the planned treatment time; technology flexibility in providing the incremental levels of difficulty needed for progressively more challenging motor learning stance phase practice; user’s ability to attempt the progression of planned challenges during stance phase motor learning; ability to show improvement in performing progressively more challenging aspects of the motor learning protocol using the technologies; users’ capability to adhere to the planned schedule of interventions, testing, and follow-up.

#### 2.4.2. User Performance Measures

We acquired user measures at baseline, after the 5th and 10th sessions and at 3-week follow-up.

*Functional mobility*. Gait speed was measured using the ten-meter walk test (TMWT [49]; MCID = 0.17 m/s [50]). The time was recorded to the nearest hundredth of a second, and three trials were averaged to provide the overall gait speed. We collected Timed Up and Go (TUG [51]; Minimal Detectable Change (MDC) = 3.53 s [52]), which assesses the speed of several motor tasks. The participants were asked to sit with their back against a chair. On the command “go”, participants were asked to rise from a chair, walk 3 m, turn, walk back and sit down. The time was recorded to the nearest hundredth of a second, and three trials were averaged. The Functional gait assessment (FGA; a maximum score of 30 points [53] and MDC = 4.2 [54]) was used to assess postural stability and coordination of motor task performance during functional walking tasks.

*Measures assessing underlying factors of stance phase control and coordination of joint movements*. First, the FM_LE_ is a measure of lower limb joint movement coordination. It has a maximum score of 34 points [55] and a Minimal Clinically Important Difference (MCID) of 6 points. Second, changes in paretic limb stance control were assessed by the treating therapist. The therapist initially assessed paretic limb stance control capability during static standing, lateral weight shifting, stride position weight shifting, slow-motion gait with upper limb support and focused cuing, and chosen speed walking. This initial assessment array was used to determine the appropriate challenge point for the initiation of the training. Subsequently, at each visit, the treating therapist conducted the same assessment in order to progress the user in stance phase practice for both in-person sessions and the home exercise program.

### 2.5. Statistical Analysis

Longitudinal linear mixed-effects models were fit to model the changes in clinical outcome measures from baseline, at the midpoint, post-treatment, and at the follow-up. Time was considered as a categorical fixed effect, and the same user-level random intercepts were included. The longitudinal models also included a covariate adjustment for the corresponding baseline value. A serial correlation was reflected either through unstructured or autoregressive order 1 covariance models. Covariance model selection was based on the model fit statistic—2 Res Log. The two-sided Type I error level was set at 0.05. The reported *p*-values of the regression model estimates of change from baseline at all study time points were adjusted for multiple comparisons using the Holm-Bonferroni correction. We also report mean estimates for change (Mean change) and Confidence Interval (CI). Analyses were performed using SAS Software (SAS Institute, Inc., Version 9.4, Cary, NC, USA).

## 3. Results

### 3.1. System User Baseline Characteristics

For the initial testing of the new stance phase training protocol, we enrolled five individuals (100% male) with first-ever stroke (>6 months post). The five individuals were aged 58.6 ± 7.8 years old (mean ± SD), Table 1), with a baseline FMLE ranging between 17 and 28 (FMLE 21.2 ± 4.5 (mean ± SD)) (Table 2). Baseline values of the other functional measures are summarized in Table 2.

### 3.2. The Protocol Feasibility

Table 3 provides the results of the feasibility factors employed. The results indicate that for the stroke survivor users in this study, the protocol was safely and feasibly administered.

### 3.3. Modelling of tDCS-Induced Electric Field

To evaluate the distribution of tDCS-induced EF using our montage and current intensity, tDCS-induced EF was modeled for each participant. As shown in Figure 2, patterns of EF distribution across the whole brain varied in each participant due to variability in stroke lesions and brain anatomy. The mean EF in the primary leg motor area was 0.17 V/m ranging from 0.13 to 0.24 V/m.

### 3.4. Functional Mobility Improvement

*Gait Speed*. For gait speed (Figure 3a) according to the longitudinal linear mixed-effects model results, a statistically significant improvement from baseline was observed after 10 treatment sessions (mean change = 0.25 m/s, 95%CI: 0.11, 0.40; *p* = 0.01). This improvement was maintained at follow-up (mean change = 0.27 m/s, 95%CI: 0.12, 0.42; *p* = 0.01). Figure 3a shows the line through means of change from the baseline for gait speed and the individual changes for each participant. The Appendix A shows the baseline and post-treatment scores for each subject.

*Timed Up and Go*. For TUG (Figure 3b), according to the regression analysis, a statistically significant improvement was observed after the 10 sessions (mean changes = −2.36 s, 95%CI: −3.81, −0.91; *p* = 0.02) and was maintained at the follow-up (mean changes = −3.24 s, 95%CI: −4.79, −1.68; *p* = 0.005). Figure 3b shows the line through means of change from the baseline for TUG and the individual changes for each participant. The Appendix A shows the baseline and post-treatment scores for each subject.

*Functional Gait Assessment*. For FGA (Figure 3c), according to the regression analysis, there was a statistically significant improvement at the midpoint (mean changes = 2.95 points, 95%CI: 1.67, 4.23; *p* = 0.003) which was maintained at the end of the 10-session treatment (mean change = 4.55 points, 95%CI: 3.08, 6.02; *p* = 0.0006) and at the follow-up (mean change = 4.38 points, 95% CI: 2.59, 6.16; *p* = 0.002). Figure 3c shows the line through means of change from the baseline for FGA and individual changes for each participant. The Appendix A shows the baseline and post-treatment scores for each subject.

### 3.5. Measures Assessing Underlying Factors of Stance Phase Control and Coordination of Joint Movements

*Fugl-Meyer limb joint coordination measure*. For FM_LE_ (Figure 4), according to the regression analysis, the FM_LE_ score improved compared to baseline as early as the midpoint assessment (Figure 4. mean change = 3.08 points, 95%CI: 0.82, 5.34; *p* = 0.04). Gains were maintained after 10 treatment sessions (mean change = 4.08 points, 95%CI: 1.82, 6.34; *p* = 0.01) and at follow-up (mean change = 4.59 points, 95%CI: 2.03, 7.15; *p* = 0.01). Figure 4 shows the line through means of change from the baseline for FM_LE_ and individual changes for each participant. The Appendix A shows the baseline and post-treatment scores for each subject.

*Changes in Paretic Limb Stance Control in Individual*. The training across all treatment modalities (VR treadmill + tDCS, overground practice, and HEP) emphasized weight acceptance and stabilization on the paretic limb. Particular emphasis was placed on achieving as-close-to-normal alignment during weight bearing on the paretic limb with as little arm support as possible. Table 4 provides descriptions of individual users’ improvements in stance phase control at the end of 10 treatment sessions.

## 4. Discussion

The results of the study contribute to the field in three ways. First, we constructed a novel protocol of gait training with brain stimulation, using bihemispheric brain tDCS framed within motor learning principles to present stance phase weight-shift coordination tasks. Overground stance phase motor learning was included in this combined protocol. We found that this combination brain-stimulation protocol was feasible for five users with chronic stroke and severe stance phase dyscoordination in terms of user safety, comfort, compliance, and endurance. Second, we found that the combined technologies feasibly provided the flexibility required for custom, precision, and the progression of the stance phase task challenge. Third, this protocol of 10-sessions of brain stimulation and motor learning-based gait training in a VR environment produced statistically and/or clinically meaningful gains in functional mobility comparable to the gains reported in longer studies [56,57,58]. That is, there was an improvement in gait speed, Timed Up and Go, and FGA, and these improvements were maintained at a three-week follow-up. However, because only five users tested the new protocol, the results cannot be generalized to the whole population.

### 4.1. Non-Invasive Brain Stimulation Can Enhance Brain Excitability and Can Be Feasibly Administered to Stroke Survivors within the Framework of A Stance Phase Coordination, Motor Learning Protocol

There is a large body of literature describing gait training methods for stroke survivors using peripherally applied methods [14,17,34,58,59,60]. Though promising in many respects, these studies did not restore normal gait coordination. Nor did they target the brain pathology that is the source of the impaired gait coordination. It is reasonable to consider that more efficacious and complete results will occur from targeting the source of deficit, in this case, the brain, as opposed to simply targeting the peripheral systems. In fact, others have demonstrated that tDCS modulates motor performance for individuals with motor deficits after stroke [24,27,31,61]. Computational models used to predict the tDCS-induced distribution of the EF in the brain revealed that the spread of current is directly dependent on the size and placement of the electrodes as well as the amount of applied current [24,46,47]. Here, we adapted the basic bihemispheric stimulation model based on the premise that using two electrodes will guide neuroplasticity by modulating the excitability of both hemispheres [23,24,38,62,63]. The results of the EF modeling for the study participants showed that this bihemispheric stimulation produced a distribution of the current near the leg M1 region (outlined in white in Figure 2), and the intensity of the field is comparable to what has been reported by other studies [64,65,66,67,68]. For each user, the leg M1 region is the optimal site in the motor cortex for evoking 50 μV MEPs in the TA muscles using the lowest TMS intensity [45]. While the predicted EFs were sensitive to anatomical diversity of the brains between participants, the overall current flow pattern was comparable across individuals, with minimal variability in the EF in leg M1 regions (Figure 2; [69,70]). In future studies on a larger population, EF estimates can be utilized for predicting clinical outcomes.

A few prior studies applied tDCS during lower limb movements in individuals with stroke [21,27,28,31,71]. They reported that tDCS modulated excitability in the lower limb motor output tracts in acute [21] and chronic stroke [31,39]. Additionally, these studies reported that tDCS enhanced gait speed after treatment in the subacute [71] and chronic stroke survivors [27,28], but there was no report of stance phase coordination recovery, and gait speed did not return to normal. To our knowledge, there are no studies that have applied tDCS during the practice of a highly focused stance phase motor coordination task that was progressed according to a motor learning-based gait training paradigm. We found it feasible for five chronic stroke survivors to participate in this protocol of combined brain stimulation and stance phase motor task practice in terms of the user-centered measures of comfort, satisfaction with utilizing the combined technologies, ability to engage in the assigned stance phase coordination training tasks, endurance, compliance with the planned number of interventions and testing procedures, and safety.

### 4.2. Combined Intervention Using Technologies and Motor Learning Principles Provided a Finely Incrementalized Method for Stance Phase Practice

Stance phase biomechanics and coordination include the complex control of the torso, pelvis, hip, knee, and ankle during the dynamic movements of weight shift laterally and forward, as well as during the progression of the body center of mass across the foot, from the heel, mid-foot, to forefoot [72]. Users in the current study exhibited dyscoordination during the gait stance phase. The exercises were incrementally progressed using a hierarchy of challenges described elsewhere [3,73,74]. Technologies provided an effective array of practice domains which we manipulated to provide the most finely-incrementalized practice possible within known motor learning principles and an enhanced learning environment. First, tDCS is a promising tool for modulating neural activities, which can improve attention and speed of processing [22,23,75,76] which, in turn, can improve learning [77,78]. Second, the VR environment provided an inviting outdoor country scene and walking path that was coordinated with the treadmill and the user stepping speed, resulting in a calm, pleasant overall environment conducive to concentration and focused attention during gait training. Third, the VR environment provided a series of obstacles in the path of the uninvolved limb, requiring a step over the obstacle with the uninvolved limb. These obstacles, then, demanded a weight shift onto the involved limb for stance phase practice. This VR-based task with a harness provided a safe method for presenting a stance phase weight shift task challenge. Further, it served to visually focus attention and demanded specific timing and a specific amount of weight bearing on the involved stance limb to accomplish the task. There were three domains of challenge that the treating therapist could modulate to provide a very specific progression of difficulty for a good motor learning response. These three domains of challenge were as follows: the speed of the treadmill (stepping speed required), the distance between VR obstacle presentation, and the height of the VR obstacle (demanding more or less weight shift to the involved (limb). An example (Table 3. ** key) of the treadmill training speed can be given for S3 and S5. S3 and S5 began gait training practice at a treadmill speed of 0.20 and 0.14 m/s, respectively. They were progressed in practice as stance limb control improved; they completed the 10th session with a treadmill gait practice speed of 0.36 and 0.28, respectively, with improved stance limb control. This is an increase of 0.16 m/s in terms of their gait practice speed with improved control. An increase of 0.16 m/s training speed meets the minimal clinically important difference (MCID) threshold for a meaningful change in training gait speed [79], at which these two users showed an improvement in their stance limb control, according to their individual training assessments (Section 3.5). In another example of improvement, by the 10th session, several users were able to perform lateral and stride position weight shifting with a controlled knee, improving from the baseline, at which time they were not able to control the knee when performing these challenges. Furthermore, overground training reinforced the VR-assisted ML practice. The overground practice began with standing in double limb support and shifting weight to the paretic limb with pelvis/hip/knee/ankle control as the coordination task. All of these tasks were finely incrementalized to ensue the progression of practice difficulty. With additional practice time, it is reasonable to anticipate a further improvement in stance phase control, which has proved, to date, difficult to treat for many in the chronic stage after stroke. With a combined protocol such as the current one, it is not possible to assign a cause of improved stance phase control to a specific portion of the protocol. In fact, it may require all of the components of this protocol to achieve the recovery of stance phase limb control. Future studies will be required to ascertain the answer to that question.

### 4.3. Combined with Brain Stimulation, Dynamic Stance Phase Motor Exercises Showed Statistically Significant Improvement in Functional Mobility

Others have reported on the effects of obstacle clearance training in chronic stroke survivors, for which VR-based training resulted in greater improvement than overground obstacle clearance training on the fastest gait speed [80]. Additional changes in gait endurance, self-selected walking speed, and stride length were noted in response to the six-session training protocol regardless of the training environment [80]. While the obstacles were adjustable, the training did not specifically target a particular gait phase (i.e., the participants were asked to clear the obstacle with whatever foot they chose) [80], nor was brain stimulation used. The mean change in gait speed in the Jaffe study [80] was 0.15 m/s across the 20 participants; in contrast, in our study, we saw an improvement of 0.25 m/s. It may be that the emphasis on paretic stance phase control in the current study led to greater improvements in gait speed response. Although, the greater improvement in the current study could be a result of the more comprehensive protocol in the current work. Both studies taken together suggest that VR-based obstacle clearance training may be a beneficial therapy for post-stroke gait training.

Gait speed improved by 0.25 m/s, which was both statistically and clinically significant [81]. Though MCID values for the FGA have not been established in chronic stroke, performance improved above the established minimal detectable change values indicating that real change occurred [54,82]. Our cohort of individuals had longstanding gait deficits from stroke and was not actively receiving conventional rehabilitation; however, even with a short duration of therapy, a clinically important change was observed. This is a preliminary work with a small cohort, testing a new protocol, but these findings are promising, and further investigation in a larger study is warranted.

### 4.4. Study Limitations

This is a protocol development study with a cohort of five users; therefore, the standard cautions should be considered when interpreting the user outcome measures. For example, the results cannot be credibly generalized to the larger population of stroke survivors because the study sample size is small, and all the participants are male. However, when testing a new protocol that coordinates multiple technologies, it is wise to begin testing with a user group such as presented in the current work. Additionally, the novel protocol was composed of multiple components, so that response to treatment cannot be pinpointed to one component. However, the body of literature that has studied one narrow intervention has not, to date, proven efficacious in restoring normal stance phase coordination to stroke survivors with persistent deficits.

## 5. Conclusions

It was feasible to administer tDCS simultaneously with highly focused ML-based VR treadmill gait training for individuals with chronic stance phase deficits after stroke. The protocol offered important methods to incrementally progress the challenge of the practice of stance phase tasks across a hierarchy of difficulty for stance phase motor tasks. For this relatively short treatment protocol, we observed statistically and/or clinically significant improvements in gait function across an array of gait performance measures which persisted at follow-up. In the future, it is important to test the efficacy of this combination intervention protocol in a randomized controlled trial for chronic stroke survivors with stance phase deficits. If proven effective in future studies, this training paradigm can easily be translated into clinical practice.

## Figures and Tables

**Figure 1 brainsci-12-00701-f001:**
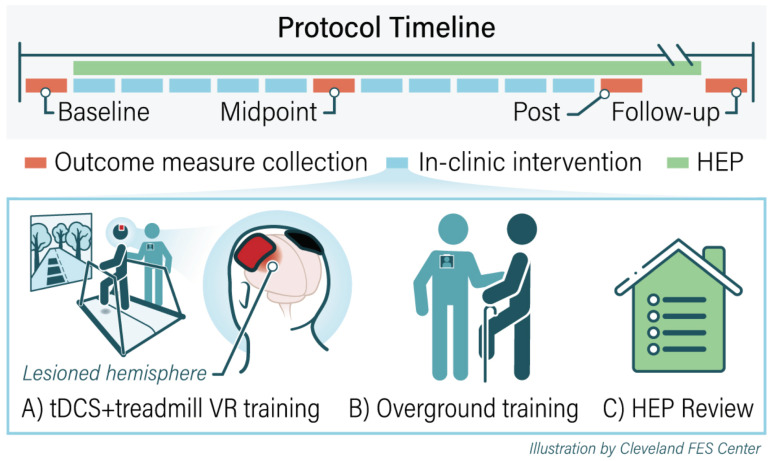
Gait training paradigm protocol. The protocol begins with baseline testing followed by 10 in-clinic intervention sessions. (**A**) During the in-clinic sessions, users practice weight shifting and targeted stance phase training using treadmill in a VR environment while stimulated by tDCS. (**B**) During the second portion of each training session, they perform task-oriented overground gait training. (**C**) At the end of the in-clinic sessions users review their personalized home exercise program (HEP). Users are asked to practice HEP independently in a safe environment at home. The users underwent testing after session 5 (midpoint), session 10 (post testing), and 3-week follow-up. users are asked to practice HEP independently during the follow-up period.

**Figure 2 brainsci-12-00701-f002:**
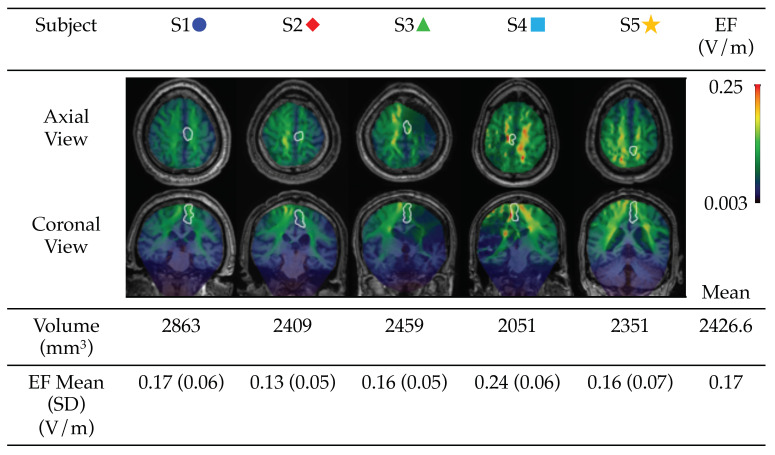
tDCS Electric Field Model. Representative axial and coronal slices showing the electric field (EF) near the leg M1 region (outlined in white) for each user. Within the M1 region, the volume and mean EF are shown.

**Figure 3 brainsci-12-00701-f003:**
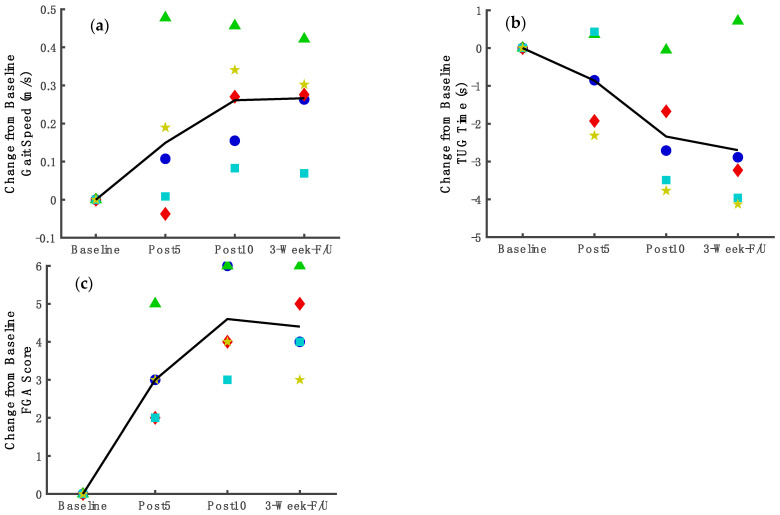
Trajectories of improvement in functional mobility and motor impairment measures throughout the study duration.

**Figure 4 brainsci-12-00701-f004:**
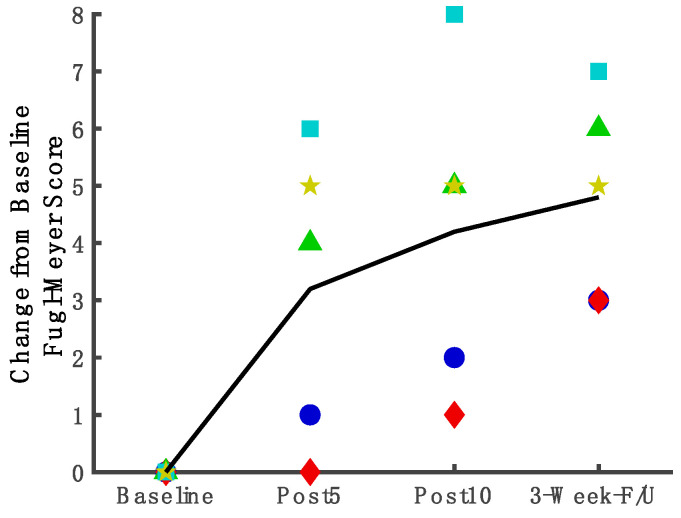
Trajectories of improvement in motor impairment measures throughout the study duration.

**Table 1 brainsci-12-00701-t001:** Participants characteristics.

User	Age	Gender	Months Post Stroke	Lesioned Hemisphere	Stroke Type
S1 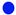	58	Male	64	Left	Hemorrhagic
S2 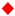	55	Male	36	Left	Hemorrhagic
S3 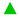	56	Male	158	Left	Ischemic
S4 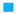	52	Male	21	Right	Ischemic
S5 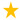	72	Male	61	Left	Ischemic

Each participant’s data are labeled consistently in all tables and figures.

**Table 2 brainsci-12-00701-t002:** Baseline Functional Measures.

User	Fugl-Meyer(Points)	Gait Speed (m/s)	TUG(s)	FGA(Points)
S1 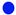	28	1.03	11.50	17
S2 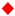	22	0.96	13.61	14
S3 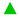	17	1.16	7.73	8
S4 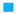	17	0.39	27.24	11
S5 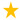	22	1.11	11.92	16
Mean (SD)	21.2 (4.5)	0.93 (0.31)	14.40 (7.49)	13.2 (3.7)

**Table 3 brainsci-12-00701-t003:** Feasibility factors.

Feasibility Factors	Findings
User (stroke survivors) recruitment	Eight stroke survivor users were evaluated for participation in the study. Five met the inclusion criteria and completed the study.
Retention for the planned duration of the treatment and testing time	All five enrolled users completed all treatment and testing sessions
User preparation and donning time of equipment	Equipment donning required an average of 15 min.
Reported comfort of the system	Users reported no discomfort regarding tDCS electrode placement or current intensity. Users reported no discomfort with the harness system or visual/auditory features of the VR.
Capability to wear and use the technologies.	All five users were able to wear the technologies and reported engagement in the presented motor tasks during tDCS and VR/treadmill training.
User endurance of the technologies for the planned treatment time	All five users demonstrated endurance for the length of each treatment session. We found it important to offer rest periods between walking practice trials on the treadmill and during phase two overground practice. Users performed a mean of 191.8 ± 32.8 (SD) obstacle-clearance steps per VR treadmill training session.
Technology flexibility in providing the incremental levels of difficulty needed for progressively more challenging motor learning stance phase practice	The technology offered numerous domains across which task difficulty could be progressed. These included treadmill speed, timing of the frequency of walking obstacles in the VR system, height of obstacles in the VR system, use of knee cage to protect joint structures during walking practice, ankle dorsi-flex assist to protect knee joint structures and assist with swing phase dorsiflexion, upper limb support, physical assist by the treating therapist, and verbal cues from the treating therapist. **
User’s ability to attempt the progression of planned challenges during stance phase motor learning	All five users showed capability to attempt task progression across one or more domains of task difficulty.
Ability to show progressive improvement in performing progressively more challenging aspects of the motor learning protocol using the technologies	All five users progressed across one or more domains of task difficulty (details provided below in the results for individual users).
Safety	All participants completed 10 training sessions with no adverse events. No users experienced a fall or near fall.

** Example of the flexible domain of treadmill training gait speeds and the recorded range from initial session to final treatment (m/s): S1: 0.18–0.24; S2: 0.38–0.38; S3: 0.2–0.35; S4: 0.20–0.16; S5: 0.14–0.28. These training speeds were separate from chosen gait speeds.

**Table 4 brainsci-12-00701-t004:** Changes in Paretic Limb Stance Control.

Users	Baseline	After Treatment
S1 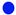	Uncontrolled/forceful knee hyperextension during weight shift/forward stepping and during mid- to terminal stance at chosen gait speed	Knee maintained in neutral position (no hyperextension) during weight shift/forward stepping at slow speed with focused attention/ upper limb support. Knee hyperextension only in very late stance with less force at chosen gait speed
S2 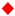	Knee hyperextension, pelvic retraction, +Trendelenberg sign * and forward flexed trunk when attempting to bear weight on the paretic limb.	Improved alignment of trunk/pelvis/hip/knee/ankle during weight shift and stepping practice; controlling knee in neutral with upper limb support while shifting weight onto paretic limb.
S3 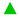	Severe/forceful knee hyperextension, +Trendelenberg sign, and pelvic retraction when attempting to bear weight on the paretic limb.	Able to protect the knee joint from excessive hyperextension forces during weight bearing by working on stance with the knee in 10° flexion. Improved alignment of paretic pelvis/hip/knee/ankle during forward weight shift to midstance.
S4 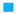	Knee hyperextension, pelvic retraction with hip external rotation, +Trendelenberg sign with center of mass between quad cane, and non-paretic limb. Step-to gait pattern with decreased time in single limb support on paretic limb.	Improved alignment of paretic pelvis/hip/knee/ankle during weight bearing with upper limb support. During chosen speed walking, taking longer steps with uninvolved limb, and pelvic retraction lessened.
S5 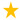	Knee hyperflexion,+Trendeleberg sign, with center of mass maintained between the non-paretic limb and quad cane.	Improved alignment of paretic pelvis/hip/knee/ankle during weight bearing with upper limb support, demonstrating knee control at neutral. During chosen speed walking, exhibited improved knee position (less flexion) during stance phase.

* +Trendelberg sign = hip abductor weakness (gluteal muscles) in the weightbearing limb (paretic limb) which results in a drop of the swinging limb pelvis (non-paretic limb) in the coronal plane.

## Data Availability

The data presented in this study are available on request from the corresponding author. The data are not publicly available due to the data are part of an on-going clinical trial.

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
