# Peer review of "Stance Phase Gait Training Post Stroke Using Simultaneous Transcranial Direct Current Stimulation and Motor Learning-Based Virtual Reality-Assisted Therapy: Protocol Development and Initial Testing"

_brainsci, 2022, doi:10.3390/brainsci12060701_

Round 1

Reviewer 1 Report

Thank you for submitting a revised version of the manuscript. I have a few minor comments:
Page 3 Line 142-144: Please cite a journal article related to stroke and tDCS research for justifying the stimulation parameter used.

Please state how the results presented in table 4 was assessed in the methods section.

As the data was only collected from 5 participants and there was no control group, please be cautious to conclude statistically significant and clinically meaningful difference.

Reviewer 2 Report

Reviews

  1. Introduction

 The introduction is nicely written. It is clear that tDCS is used with virtual reality to improve gait training methods. The stance phase is chosen due to the issue that stroke survivor faces during this phase.

 Methods

  • In the method section, the schematic for the experimental design presented shows the midpoint, follow-up, etc. It will be good if labels showing the week are also added. It will be clear to the reader how many weeks are spent on training and follow-up.

 Results

 Results are presented nicely.

  • However, based on your claim in the introduction that your protocol is novel and likely better than other protocols that use the peripheral nervous system for rehab therapy. I recommend a comparison between your protocol and other protocol, how does your protocol surpass other rehab protocols? If you can present the data (weekly recovery of your protocol vs others). It will be great and will support your claim in the introduction further that your protocol stands better than others.

 Discussion

  • In discussion, can you clearly define what is leg M1 region is or provide a reference for that? The M1 region of the brain to my understanding is not very modular so it is hard to classify the M1 region that controls the leg. (Line 436)
  • I suggest citing the following literature for the statement about the stance phase biomechanics Line (461-464). (Singh, R. E., Iqbal, K., and White, G., 2020).
  • Overall, the discussion is written nicely and extensively. I also recommend in your limitation; that you should suggest the study is gender-biased. There were no female participants in the study.
  1. Conclusion

 No comments on the conclusion

Overall, the paper is nicely written. The study protocol is well discussed. Results are presented nicely, and literature has been discussed extensively. I have suggested some more addition in the results section. In addition, I strongly encourage the authors to mention in the title that this study accounts for only male patients in the study. Thus, rephrasing the title as below. Note: This is an example of your title choice you can rephrase it yourself.

“Stance Phase Gait Training Post Stroke in male patients using simultaneous 2 transcranial Direct Current Stimulation and Motor Learning- 3 based Virtual Reality-assisted Therapy: Protocol Development 4 and Initial Testing”

References

  1. Singh, R. E., Iqbal, K., and White, G. (2020). Proficiency‐based recruitment of muscle synergies in a highly perturbed walking task (slackline). Engineering Reports.

Reviewer 3 Report

The manuscript provides a novel intervention for gait in stroke participants. it is well written and important. it should be published. the manuscript needs some revisions to be more  informative and clear. Please find the following comments: 

1- The manuscript is very long especially the abstract and discussion
2- Discuss the previous literature that examines the combined interventions for stroke.
3- Discuss the effect of your interventions in term of minimal clinical importance difference for the selected outcomes in chronic stroke patients

4- Line 421: Need references 
